# Polyphenol supplementation and executive functioning in overweight and obese adults at risk of cognitive impairment: A systematic review and meta-analysis

Sara Farag[1]*, Catherine Tsang[2], Philip N. Murphy[1]*

**1** Department of Psychology, Edge Hill University, Ormskirk, Lancashire, United Kingdom, **2** Faculty of Health and Life Sciences, Applied Sciences, Northumbria University, Newcastle-Upon-Tyne, United Kingdom

☯ These authors contributed equally to this work.
* 24058416@edgehill.ac.uk (SF); murphyp@edgehill.ac.uk (PNM)

## Abstract

### Background and objectives

Increasing evidence indicates a link between obesity and cognitive impairment. Furthermore, there is limited literature regarding the effect of polyphenols, a plant derived compounds, on executive functioning in an overweight/obese population at-risk of cognitive impairment. The aim of the present systematic review and meta-analysis of randomized controlled trials is to examine the effect of polyphenol supplementation on executive functions in overweight and/or obese populations at risk of cognitive impairment.

### Methods

A comprehensive literature search was conducted from inception to March 2023 using four electronic databases: PubMed/Medline, PsycInfo, Scopus and Cochrane trials library. Published primary research studies in English that compared the effect of polyphenols with placebo on executive function in overweight/obese adults were considered eligible for the meta-analysis. Jadad scale was used for the methodological quality rating of the included studies. Hedges $g$ with 95% confidence intervals (CI) for endpoints were calculated using random effect model where applicable. Rosenthal's Fail-safe N, funnel plots, the Begg and Mazumdar's rank correlation test (Kendall's S statistic P-Q), Egger's linear regression test, and Duval and Tweedie's trim-and-fill test were identified for potential use as appropriate, to examine publication bias. Sensitivity analysis was conducted to examine the robustness of the results.

### Results and conclusion

A total of 23 RCT studies involving $N$ = 1,976 participants were included in the review. The results of the meta-analysis revealed a non-significant effect for polyphenol supplementation on executive function ($g$ = 0.076, CI = -0.018 to 0.170). Observations from primary studies within the meta-analysis showed a potential positive effect of polyphenol

**Data Availability Statement:** All relevant data are within the paper and its Supporting Information files.

**Funding:** The authors received no specific funding for this work.

**Competing interests:** The authors have declared that no competing interests exist.

supplementation in a younger population at-risk of cognitive impairment and it is recommended to investigate this further in future studies. Moreover, the variability of the tasks used to examine executive functions as well as the adequate reporting of supplement's phenolic composition is a limitation that future work should also consider.

## Introduction

It is recognised that overweight and obesity pose an increased risk for the development of cardiometabolic disease, and increasing evidence indicates a link to cognitive impairment associated with early onset dementia in such populations [1–3]. Dementia is associated with severe cognitive impairment, reduced quality of life and mortality, and is considered a major public health crisis affecting over 50 million people globally [4]. Obesity is characterised by excessive adiposity, and increased concentrations of the pro-inflammatory adipokines tumour necrosis factor alpha (TNFα) and interleukin-6 (IL-6), leading to a chronic low-grade state of inflammation [5, 6]. An increased production of reactive oxygen species (ROS), oxidative stress, and reduced antioxidant defences have also been observed in such populations [7, 8]. The consequences of these biochemical changes include neuroinflammation, neuronal damage, gliosis (fibrosis of brain tissue) and neuronal cell death [9, 10] with the consequent possibility of cognitive impairment [11].

Relationships between impaired executive functions (EFs) and obesity/overweight have been widely reported, with neuroinflammation and oxidative stress being highlighted as risk factors [2, 3, 12, 13]. EFs are a sub-group of cognitive functions which constitute part of the working memory and decision-making processes which enable people to regulate their daily lives [14–16]. EFs are characterised by their active higher order processing of information, as opposed to more basic processes of information acquisition, retention, and retrieval [15, 17, 18]. Specific EFs have been identified empirically and by reviews of published work, and include the conscious control of attentional focus, updating responses and decisions based upon the incoming flow of information to the brain, and control over pre-potent responses (i.e., response inhibition) [15, 17–19]. Management of the visuospatial environment has been identified as an important EF [18] and supports such daily activities as driving a car and operating machinery. Despite empirical bases for some laboratory tasks to be accepted as measures of specific EFs, *ad hoc* decisions are often made by research teams and authors with regard to tasks being an EF measure as opposed to a measure of more basic cognitive functioning. Impairments to EFs clearly have the potential to diminish people's ability to effectively manage their daily lives and have been extensively researched in relation to both normal ageing and dementia [20–22], and in relation to other causal influences including sleep disorders [23], alcohol use [24], and illicit drug use [25].

The relationship between problems of neuroinflammation and oxidative stress in obese/overweight populations, and risks of EF impairment, raises the issue of interventions to address such problems. A possible intervention is the use of polyphenols, which are secondary plant metabolites widely dispersed in abundant dietary sources including fruits, vegetables, tea, coffee, and cacao [26], and which are reported to have antioxidant and anti-inflammatory properties [9, 27]. A literature review of epidemiological and randomised controlled trials (RCTs) has reported positive outcomes for polyphenols regarding cerebrovascular and neurodegenerative induced cognitive impairment, but this positive effect was reported in healthy young adults who did not have obesity as a risk factor [28]. Research with older populations

has produced mixed results. For example, polyphenol supplementation in the form of spearmint has been reported to enhance performance on working memory and visuospatial tasks [29], with the tasks used drawing upon identified EFs [17–19]. However, another study which used blueberry supplementation reported increases in both brain activation in task-associated regions and resting-state quantitative grey matter perfusion following a 12-week intervention, but without a change in EF performance [30]. Other studies have reported no effect on EF performance even where performance on more basic areas of cognitive performance did improve [31–33].

The purpose of this current systematic review and meta-analyses of RCTs using polyphenol supplementation was to investigate further the effectiveness of this intervention with obese/overweight participants. Its main outcome objective was to elaborate upon existing knowledge of the effectiveness or otherwise of polyphenols in general for the improvement of EFs in an obese/overweight population at risk of cognitive impairment. A secondary objective was to identify if some polyphenols as opposed to others, might be beneficial for EFs in this population.

## Methods

To fulfil the aim of investigating the effect of polyphenols on EFs, a research protocol (Unpublished and un-registered: S1 Table) was used and Preferred Reporting Items for Systematic Reviews and Meta-analysis (PRISMA) guidelines were followed in the present study [34].

### Search strategy

The PubMed-Medline, PsycInfo, Scopus and Cochrane trials library databases were systematically searched for RCTs published in the English language from inception until March 2023 using key terms containing both a polyphenol component and a functioning component. The polyphenol search terms included 15 keywords which were polyphenol, pomegranate, flavonoids, polyphenolic compound, polyphenolic compounds, isoflavone, flavanol, phytoestrogen, resveratrol, ellagitannin, ellagic acid, punicalagin, or anthocyanins, proanthocyanidin and proanthocyanidins. These keywords were searched independently and after that, their results were combined by the operator instruction 'OR.' The functioning component contained 32 items and were; mild cognitive impairment, MCI, cognition, cognitive performance, cognitive function, brain function, executive function, neuroimaging, neural, magnetic resonance imaging, MRI, fMRI, grey matter, gray matter, brain structure, electrophysiology, EEG, event related potential, neuroblast, cerebral blood flow, CBF, regional perfusion, pulsatility index, transcranial doppler, TCD, near-infrared spectroscopy, NIRS, total haemoglobin, oxygenated haemoglobin, oxy-Hb, deoxygenated haemoglobin and Deoxy-Hb. Similar to the polyphenol search, the results of each independent search of the cognitive keywords were combined by the operator instructor 'OR'. The polyphenol component and the functioning component were linked with the operator instruction 'AND.' The search results were then filtered by being an RCT to have more focussed search results. Items in each component were intentionally broad so as to identify as many studies as possible. Specific EFs and tests of such functions were not used as studies often evaluate composite performance factors which subsume such functions, so that they are not explicitly mentioned. Reporting in primary sources also often highlights use of recognised test batteries rather than specific tests.

Reference lists yielded by the respective databases were cross-matched, and duplicates removed manually and by RefWorks. Each publication was then evaluated in relation to the inclusion criteria, with further relevant studies being identified, if possible, from these sources.

## Study selection

For inclusion, studies were required to meet criteria concerning population, intervention, comparison, outcome, and study design (PICOS) (Table 1). additional inclusion criteria were for the studies to be primary research published in peer-reviewed journals in English. People with mild cognitive impairments (MCI) or those who complained of subjective memory problems in sample members were acceptable for inclusion.

Although studies frequently reported a range of cognitive tasks, the focus of this review led to only results from EF tasks being included in the meta-analysis. Designation as an EF task was determined by it matching at least one of the following four criteria. These were that the article explicitly reported the task to be an EF task, that it was recognised as an EF task in the wider literature, that an empirically or logically demonstrated link to EF had been reported [15, 17–19], or that its performance requirements corresponded to accounts of EF in the literature. For example, vigilance tasks were included because they required the inhibition of attention switching and, therefore, engaged two empirically demonstrated executive functions (i.e., attention switching/shifting and inhibition) [17, 19].

## Data extraction

After identifying eligible publications, data comprising, the first author's name, publication year, target population, study design, treatment characteristics, the dosage and duration of the intervention, and the outcome concerning EF performance were extracted from each study. This was then followed by the verification of these information by two independent investigators.

## Quality assessment

The Jadad scale [35], a validated, short, and reliable 3-item instrument was used to assess the quality of the included studies. Each study was reviewed by answering three main questions concerning randomisation, blinding, and withdrawal/dropouts. A consensus meeting with the authors was conducted to resolve any uncertainties raised by any of the authors regarding the quality of an article. A maximum score of 5 (the sum of awarded points) was given only when all criteria were clearly satisfied. Previous reviews [36, 37] have considered a score of 3 or more to be "high quality", with studies scoring less than 3 being considered as "low quality".

**Table 1. PICOS (population, intervention, comparison, outcome, and study design) criteria for inclusion of studies.**

| Parameter | Inclusion criteria |
|---|---|
| **Population** | Obese and/or overweight adults age>18 (Body Mass Index (BMI) $\geq$ 25kg/m2) |
| **Intervention** | Acute and/or chronic polyphenols-rich supplementation |
| **Comparator** | Any: food, juice, placebo |
| **Outcome** | Executive function tasks |
| **Study design** | Randomised controlled trials |

Exclusion criteria removed studies written in languages other than English, studies conducted in people diagnosed with severe cognitive impairment/dementia, animal and *in-vitro* studies, case studies, encyclopaedias, book chapters and reviews. Also excluded were studies where no EF tasks were administered or no supplementation was given, studies administrating multiple supplements additional to polyphenols, publications from conferences or workshops, and studies with severe methodological deficiencies such as allocation not being randomized, participants not blinded, absence of control comparison (e.g., no placebo condition), and inappropriate statistical procedures.

## Meta-analytic strategy

In all analyses, Comprehensive Meta-Analysis (CMA for Windows, Version 3, Biostat, Engle-wood, NJ 2013, USA) was used. Following consultation with external advisors, as all studies were RCTs, the data entered were post intervention scores for the intervention and placebo conditions, respectively. To preserve the integrity of the summary effect size, the mean of the multiple comparisons within a study was taken where more than one relevant dependent variable was reported [e.g., 38], where different supplementation doses were compared to a placebo or baseline condition [e.g., 39], and/or different durations since dose administration were used in comparisons [e.g., 40]. As only one value for each primary study was included in analysis, risks of distortion arising from studies with parallel arms (Between participants design) were avoided [41].

Three levels of meta-analyses were conducted. Level One comprised an overall meta-analysis of all primary studies in the sample. Level Two comprised separate analyses for Between participant RCTs (BTW-P RCT) and crossover trials as a precaution against any distortions arising from the mixing of different types of trial, as described in the Cochrane Handbook [41]. Sensitivity analyses were conducted at Level Two by removing one study at a time as described by Borenstein et al. [42]. Level Three meta-analyses maintained the distinction between BTW-P RCTs and crossover trials but were conducted on studies having administered the same polyphenol and those that failed to specify the polyphenol content of their supplementation were grouped separately.

In keeping with good practice for meta-analyses [42, 43], the Level One and Two meta-analyses used random-effects models as an *a priori* choice, as opposed to the choice of model being based upon tests for heterogeneity of effect sizes. The latter practice is strongly criticised by Borenstein et al. [43] on the grounds that the test for heterogeneity using the $Q$ statistic is characterised by low power and may consequently not be sensitive to a distribution of true population effect sizes from which the primary studies have gathered samples of data. Our Level One and Two meta-analyses comprised studies administering a range of different polyphenols, so that an *a priori* basis existed to assume a need to control for a distribution of true effect sizes to have been sampled. Consequently, the *a priori* choice of random effects models was appropriate for these analyses. For the Level Three meta-analyses, fixed-effect models were used in the first instance, since polyphenol type did not differ between studies, although the assumption of the lack of heterogeneity in true effect sizes was checked against results from the $Q$ statistic.

Hedges $g$ was used as the effect size metric for all variables to avoid distortions arising from small sample sizes in some primary studies. Outcomes were coded as positive where supplementation conditions showed better performance than control conditions, and as negative where the opposite was the case. In all analyses the alpha level was $P \leq 0.05$. In the event of the Level One meta-analysis yielding a significant result, the following tests for publication bias were identified for use: Rosenthal's fail-safe N statistics, Begg, and Mazumdar's rank correlation test (Kendall's $S$ statistic $P$-$Q$) [44], Egger's linear regression test [45], and Duval and Tweedie's trim-and-fill test [46]. Although the fail-safe $N$ test would be irrelevant if the results of the Level One analysis were not significant it would still be useful to know if the obtained summary effect size had been biased by the presence of smaller studies with larger effect sizes than those obtained from larger samples. Consequently, the remaining tests of publication bias would still be conducted.

# Results

## Study selection

Following the systematic search 4365 records were identified. Of those, 2363 articles were excluded as not being RCTs, or as being literature reviews or clinical studies that were still

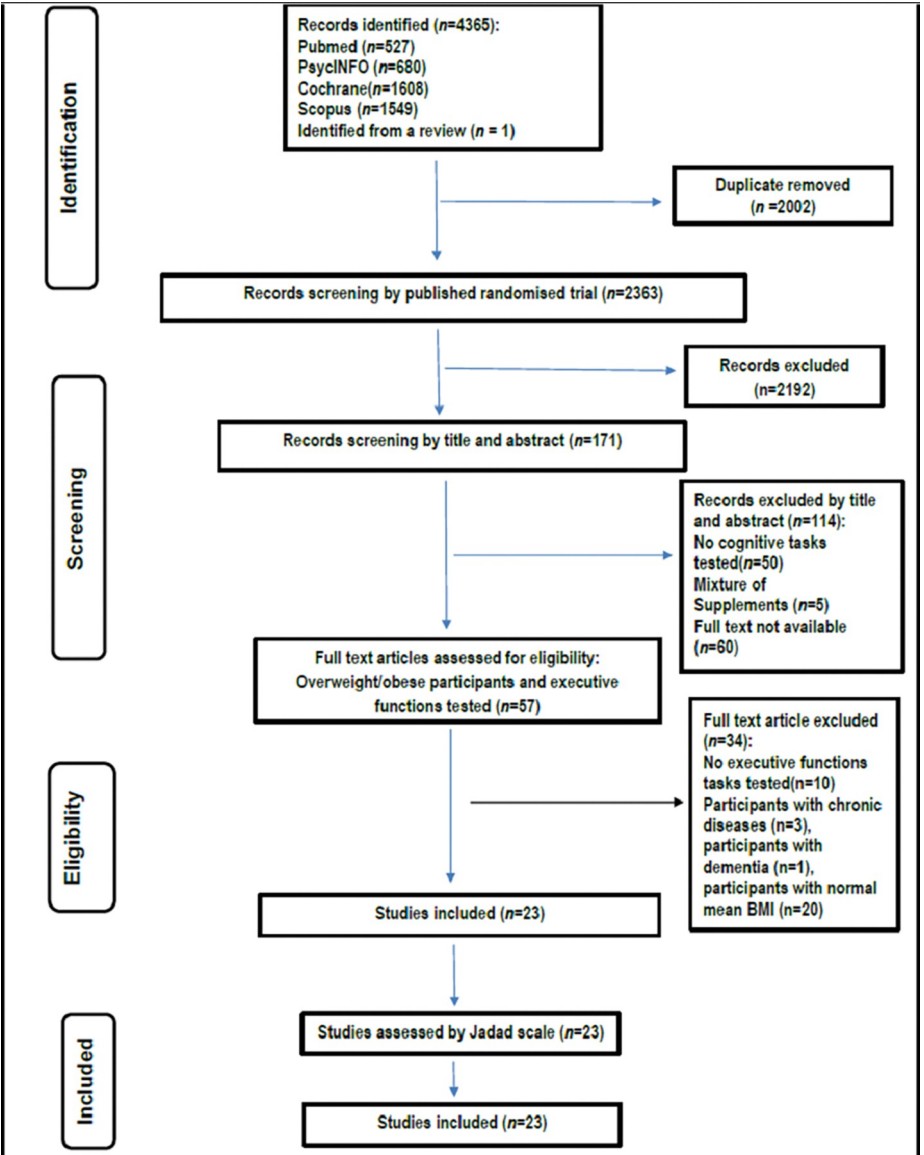

**Fig 1. Flow chart of the studies selection process.**

recruiting at the time of literature search, in addition to those which included participants with chronic diseases or who were not obese or overweight. Finally, the full text of 23 remaining studies were assessed which met the inclusion criteria and were included in the present sample. It is important to note that some studies [33, 47, 48] did not report the exact polyphenolic content of the administered supplement. The PRISMA flow chart of the study selection process is shown in (Fig 1).

## Study characteristics

Table 2 summarises the data extracted from the 23 primary studies in this review. Eight studies implemented a crossover study design while the remaining 15 studies had a between-participants design (BTW-P). While 6 studies investigated the acute effect of polyphenol

**Table 2. General characteristics of the included studies.**

| Author | Design | Intervention and dose | Phenolic content | Elapse time to cognitive testing, and washout | Study population | | Executive functions | Outcomes |
|---|---|---|---|---|---|---|---|---|
| | | | | | **Treatment group** | **Placebo group** | | |
| [32] Ahles et al. (2020) | Double blind, randomized placebo-controlled parallel study | 150 mg/day Aronia melanocarpa extract (AME) 90/day Aronia melanocarpa extract (AME) | 150 mg AME capsule contained 27 mg anthocyanin. 90 mg AME capsule contained 16 mg anthocyanis | Chronic effect tested at baseline, 6-, 12- and 24-weeks post supplementation; washout N/A | Healthy adults $N = 34$ AME 90mg group: Mean age 53±1, BMI 29.5± 0.4 kg/m² $N = 35$ AME 150 mg group: Mean age, 53 ±1, BMI 29.4 ± 0.5 kg/m² | Healthy adults $N = 32$ Mean age 53±1, BMI 29.3 ± 0.5 kg/m² | The Stroop and Number cross-out task. | No significant effect found for EFs. |
| [31] Alharbi et al. (2016) | Double blind, randomized placebo-controlled cross-over trial | 240ml Flavonoid-rich orange drink | 272 mg flavonoids/drink | Acute effects tested at baseline, 2- and 6-hours post consumption: 2-week washout between arms of the trial. | $N = 24$ Healthy males mean age 51±6.6 years. Mean BMI 28.3 ± 3.1 kg/m² | | Digit Symbol Substitution Test, Serial 7s and continuous performance task | No significant effect found for EFs. |
| [49] Anton et al. (2018) | Double blind, Phase IIa randomized placebo-controlled trial | Resveratrol | Low dose: 300 mg/day. High dose: 1000 mg/day | Chronic administration for 90 days before testing: washout N/A | Older adults $N = 12$ Low dose Mean age 73.17 ± 2.08 years, mean BMI 29.8±0.62 kg/m²: $N = 10$ High dose: mean age 73.60 ± 2.5 years, mean BMI 29.0±1 kg/m²) | Older adults $N = 10$ Mean age:73.30 ±2.06 years, mean BMI:29.7 ± 0.62 kg/m²: | Trail making test B, Digit Symbol Substitution Test, Erikson-flanker test, controlled oral word association and task switching. | No significant effect found for EFs. |
| [50] Boespflug et al. (2017) | Double blind, randomized, placebo-controlled trial | 25g/day Freeze dried, whole fruit blueberry powder | 417 gallic acid equivalents (Total polyphenols), anthocyanin:269 mg cyanidin 3-glucoside equivalents | Chronic administration for 16 weeks before testing: washout N/A | $N = 8$ Older adults with MCI mean age 80.4 ± 7.3 years, mean BMI 26.2 ±3.6 kg/m² | N = 8 Older adults with MCI: mean age 75.5 ± 4.8 years, mean BMI 26.4 ±2.4 kg/m² | Sequential letter n-back (1 and 2 back) | No significant effect found for EFs. |
| [51] Bondonno et al. (2020) | Double blind, randomized, placebo-controlled cross-over trial | Enzyme modified isoquercitrin (EMIQ®) (CC) | 4.89mg isoquercitrin | Acute administration at 4-hours: washout: N/A | $N = 23$ Adult with at least one cardiovascular disease risk factor, mean age 64±16 years old, mean BMI 27 ± 3.7 kg/m² | | Serial 3s and Serial 7s and rapid visual information processing task. | No significant effect found for EFs. |
| [30] Bowtell et al. (2017) | Double blind, randomized, placebo-controlled trial | 30ml/day of blueberry concentrate | 387 mg anthocyanidin | Chronic administration for 12 weeks before testing: washout N/A | Older adults $N = 12$ Mean age 67.5 ± 3.0 years, mean BMI 25.9 ± 3.3 kg/m² | Older adults $N = 14$ Mean age 69.0 ± 3.3 years, BMI 27.1 ± 4.0 kg/m² | Groton maze learning test and numerical Stroop while fMRI is performed, identification task and N-back (1 and 2 back) | No significant effect found for EFs. |
| [52] Cook et al. (2020) | Double blind, Randomized, placebo-controlled cross-over trial | 600mg/day New Zealand blackcurrant extract | 210 mg of anthocyanin | Acute administration for 7 days before testing: washout: 1 week | $N = 14$ Older adults mean age 69 ± 4 years, mean BMI 28.5 ± 2.9 kg/m² | | Spatial working memory and rapid visual information processing task. | No significant effect found for EFs. |

*(Continued)*

**Table 2.** (Continued)

| Author | Design | Intervention and dose | Phenolic content | Elapse time to cognitive testing, and washout | Study population | | Executive functions | Outcomes |
|---|---|---|---|---|---|---|---|---|
| | | | | | Treatment group | Placebo group | | |
| [47] Cox et al. (2014) | Double blind, randomized placebo-controlled parallel study | 80 mg Curcumin (CC) | Not mentioned | Acute (At 1,3 hours) chronic (4 weeks) acute-on-chronic (1 h and 3 h after a single dose following 4-week treatment) administration before testing: washout: N/A | Healthy adults $N = 30$ Mean age 67.56 ± 4.47 years, mean BMI 25.5 ± 3.481 kg/m$^2$ | Healthy adults $N = 30$ Mean age 69.43 ± 6.579 years, mean BMI 27.2 ± 4.818 kg/m$^2$ | Digit vigilance task and serial 3s and subtraction task | Significant benefit of curcumin over placebo at 1-hour for Digit vigilance and serial 3s subtraction task and serial 3s at 4-weeks |
| [38] Dodd et al. (2019) | Single-blind randomised, controlled, cross-over trial | 30g (dissolved in water) blueberry beverage | 579 mg of antho-cyanidins and pro-cyanidins | Acute effects on recall tested at baseline and at 2- and 5-hours post consumption: washout not reported | $N = 18$ Adults mean age: 68.7 ± 3.3 years, BMI 25.8±4.46 kg/m$^2$ | | Go-NoGo, the Stroop test, digit switch, digit symbol substitution test, continuous performance task, random word generation, three-word sets task, n-back, letter memory and location task, | No significant effect found in EFs. However, a trend towards reduced switch cost following the blueberry drink compared to the control at the 2-hour testing point. |
| [53] Evans et al. (2017) | Double blind, randomized placebo-controlled parallel comparison dietary intervention trial | 75 mg twice /day Resveratrol | 99% pure synthetic trans-resveratrol (ResVida™) | Chronic administration for 14 weeks before testing: washout: N/A | Healthy postmenopausal women $N = 38$ Resveratrol mean age 61.5 ± 1.1 years, BMI 26.8 ± 0.82 kg/m$^2$ | Healthy postmenopausal women $N = 41$ Placebo mean age 61.5 ± 1.2 years, BMI 26.6 ± 0.8 kg/m$^2$ | Visuospatial Working Memory double span and TMT B | Significant effect for the TMT B number of errors |
| [39] Fournier et al. (2007) | Double blind, randomized, placebo-controlled trial | 353 ml/day soy milk 70 mg isoflavones/ day supplement | Soya milk: isoflavones 71.6mg Soya supplément; isoflavones 70 mg | Chronic administration for 16 weeks before testing: washout N/A | Healthy postmenopausal women $N = 25$ Soy milk mean age 56.1±0.9 years, BMI 26.8 ±1.2 kg/m$^2$ $N = 27$ isoflavone supplement mean age 55.7 ± 0.7years, BMI 28.2 ± 0.9 kg/m$^2$) | Healthy postmenopausal women $N = 27$ Mean age 56.4 ±0.8 years, mean BMI 28.5 ± 1.3 kg/m$^2$: | The Stroop task, corsi block-tapping task and colour matching task | No significant effect found for EFs. |
| [54] Henderson et al. (2012) | Double blind, randomized placebo-controlled parallel study | 25 g/day Isoflavone-rich soy protein | 91 mg aglycone weight isoflavones | Chronic administration for 2.5 years before testing: washout N/A | Healthy postmenopausal women $N = 154$ Mean age 61±7 years, mean BMI 26.5 ± 5 kg/m$^2$) | Healthy postmenopausal women $N = 159$ Mean age 61 ± 7 years, BMI 26.7± 5 kg/m$^2$: | Trial making task B, block design and category fluency, Boston naming test | No significant effect found for EFs. |

(*Continued*)

**Table 2.** (Continued)

| Author | Design | Intervention and dose | Phenolic content | Elapse time to cognitive testing, and washout | Study population | | Executive functions | Outcomes |
|---|---|---|---|---|---|---|---|---|
| | | | | | Treatment group | Placebo group | | |
| [29] Herrlinger et al. (2018) | Double blind, randomized placebo-controlled parallel study | 600mg (14.5% rosmarnic acid and 24% total polyphenols) Spearmint extract 900mg (14.5% rosmarnic acid and 24% total polyphenols) Spearmint extract | Spearmint extract;14.5% rosmarnic acid and 24% total polyphenols | Chronic administration 90 days before testing: washout: N/A | Healthy participants with age-associated memory impairment $N = 30$ Spearmint extract (600 mg group) Mean age 59.1 ±1.0 years, BMI 27.1 ± 0.7 kg/m² $N = 30$ Spearmint extract (900mg group) mean age 60.8 ±1.0 years, mean BMI 27.9 ± 0.7 kg/m²) | Healthy participants with age-associated memory impairment $N = 30$ Mean age 58.2 ±1.2 years, BMI = 25.9 ± 0.7 kg/m² | Spatial and numeric working memory, | Significant improvement in accuracy of spatial working memory and quality of working memory after supplementation with 900 mg spearmint extract. |
| [55] Huhn et al. (2018) | Double blind, randomized placebo-controlled parallel study | 200 mg resveratrol and 320 mg quercetin/day | Resveratrol and quercetin | Chronic administration for 26 weeks before testing: washout N/A | Healthy adults $N = 27$ Mean age 68.60 ± 4.92 years, mean BMI 26.5 ± 3.8 kg/m²) | Healthy adults $N = 26$ Mean age 67.54 ± 5.07 years, mean BMI 26.9 ± 4.6 kg/m², | Trial making task B | No significant effect found in any parameter. |
| [56] Igwea et al. (2020) | Randomized crossover clinical trial | 200 mL/day of Plum nectar | Anthocyanin (mg Cyanidin-3-glucoside equivalents) 3.7–5.3/100ml | Chronic administration for 8 weeks before testing: washout: 6 week | $N = 28$ Healthy old adults mean age 70 ± 10 years, mean BMI 26 ± 4 kg/m² | | Letter and category fluency Stroop | No significant effect found for EFs. |
| [57] Keane et al. (2016) | Double blind, crossover randomized Placebo controlled trial (Latin-square design) | 60 ml Montmorency tart cherries concentrate | 68·0 mg cyanidin-3-glucoside/l, 160·75 gallic acid equivalent/l | Acute administration. Testing done 1, 2, 3 and 5 h after consumption. Washout: 14 days | $N = 27$ Healthy adults mean age 50± 6years, mean BMI 26.1 ± 4.9kg/m² | | Digit vigilance Rapid visual information processing Stroop | No significant effect found for EFs. |
| [40] Kennedy et al. (2017) | Double blind, crossover randomized Placebo controlled trial | 800 Green-oats extract 1600 mg Green-oats extract (GOE) | Flavonoid content (calculated as isovitexin of ≥0.3% (w/w) | Acute effects on recall tested at baseline, and at 1, 2.5, 4- and 6-hours post-consumption: 1-week washout between arms of the trial. | $N = 42$ Healthy participants with subjective memory complaint mean age: 58.90 ± 4.8 years, BMI 25.5 ± 3.18 kg/m² | | Peg and Ball task, Stroop task, Corsi blocks task and digit vigilance task. | A significant main effect of treatment on the speed of performance measure across all tasks, thinking time and completion time for the Peg and Ball task following 800 mg dose. Also, a significant treatment × occasion interaction for the % maximum span score on the Corsi Blocks following 800 mg GOE during second visit only. |

*(Continued)*

**Table 2.** (Continued)

| Author | Design | Intervention and dose | Phenolic content | Elapse time to cognitive testing, and washout | Study population | | Executive functions | Outcomes |
|---|---|---|---|---|---|---|---|---|
| | | | | | Treatment group | Placebo group | | |
| [58] Kreijkamp-Kaspers et al. (2004) | Double-blind, randomized placebo-controlled trial | 25.6 g/day Isoflavone-rich soy protein | 52 mg genistein, 41 mg daidzein, and 6 mg glycitein | Chronic administration for 12 months before testing: washout N/A | Healthy postmenopausal women $N$ = 88 Mean age 66.5 ± 4.7years, BMI 26.4 ± 4.1 kg/m$^2$ | Healthy postmenopausal women $N$ = 87 Mean age 66.7 ± 4.8 years, BMI 25.9 ± 3.5 kg/m$^2$ | Digit symbol substitution Category fluency, Boston naming task and Trial making task B. | No significant effect found in any parameter. |
| [48] Krikorian et al. (2022) | Double-blind, randomized placebo-controlled trial | 1 packet powder/day freeze dried blueberry equivalent to 0.5 cc whole-fruit | Not mentioned | Chronic administration for 12 weeks before testing: washout: N/A | Healthy adults $N$ = 13 Mean age 55.60 years, mean BMI 31.7 kg/m$^2$ | Healthy adults $N$ = 14 Mean age 57.2 years, mean BMI 33.2 kg/m$^2$ | Controlled Oral Word Association Test | Significant effect for the BB group for phonemic not the category form. |
| [33] Sala-Vila et al. (2020) | Parallel-group, observer-blinded, randomized controlled trial | 30-60g/day Diet enriched with walnuts | Not mentioned | Chronic administration for 24 months before testing: washout N/A | Healthy elders $N$ = 336 Mean age 69.4 (CI 69.0, 69.8) years, mean BMI 27.1 (CI 26.7, 27.6) kg/m$^2$ | Healthy elders $N$ = 321 Mean age 68.9 (CI: 68.5, 69.3) years, mean BM 27.4 (CI 27.0, 27.9) kg/m$^2$ | Stroop Colour naming trial, Boston naming task, Block design and Trial making task B. | No significant effect found for EFs. |
| [59] Wong et al. (2013) | Double blind, crossover randomized Placebo controlled trial | 75mg Resveratrol | 99% pure synthetic trans-resveratrol | Chronic administration for 12 weeks before testing: washout: 1 week | $N$ = 28 Obese Males and post-menopausal females mean age 61 ± 1.3 years, mean BMI 33.3 ± 0.6 kg/m$^2$ | | Stroop colour-word test | No significant effect found for EFs. |
| [60] Yahya et al. (2017) | Double blind, randomized placebo controlled trial | Two 250mg/day capsule Polygonum minus extract | Quercetin-3-glucuronide and Quercitrin (04% and 0.1% respectively) | Chronic administration for 6 weeks. Testing at 3 and 6 weeks: washout: N/A | Healthy middle-aged women N = 17 Mean age 44.7 ± 4.6 years, mean BMI 28.2 ± 5.0 kg/m$^2$ | Healthy middle-aged women N = 18 Mean age 45.6 ±6.1 years, mean BMI 27.7 ± 3.9kg/m$^2$ | Comprehensive trail making test | No significant effect found for EFs |
| [61] You et al. (2021) | Double blind randomized placebo-controlled trial | Two 250 mg capsules Cosmos caudatus (CC)/day | 7.41 Total phenolic content (mg Gallic acid equivalent) (Quercetin 0.9 and Quercitrin 1 (%w/w)/100g) | Chronic administration for 12 weeks before testing: washout: N/A | Old adults with MCI $N$ = 10 Mean age 64± 4.0 years, mean BMI 26.1 ± 3.19 kg/m$^2$ | Old adults with MCI $N$ = 10 Mean age 63.40 ±2.41 years, mean BMI 25.9 ± 2.7 kg/m$^2$ | N-Back and Stroop word colour task | No significant effect found for EFs. |

Placebo (PLA), Commercial company (CC), Aronia melanocarpa extract (AME), Green-oats extract (GOE), weight per weight (w/w), Mild cognitive impairment (MCI), Trail Making Test (TMT), Functional magnetic resonance imaging(fMRI), Body mass index (BMI), not applicable (N/A)

supplementation, 1 examined the acute, chronic, and acute versus chronic effect of polyphenols, and the remaining 16 studies investigated the chronic effect of polyphenol supplementation.

**Participant characteristics.** In total, 1,976 individual participants were included for analysis. The sample size varied in the primary studies employing a crossover design, from 14 [52] to 42 [40], whilst studies with a between-participants design ranged from 16 [46] to 657 participants [33]. The mean of the reported mean ages of participants in all 23 studies receiving polyphenol supplementation was 62.92 years (SD = 8.06 years). For the 15 studies with between-

participant designs the mean age for the supplementation groups was 62.51 years ($SD = 8.75$ years) compared to 62.90 years ($SD = 7.97$ years) for the control groups. Mean BMIs ranged from 25.5 kg/m$^2$ [47] to 33.7 kg/m$^2$ [59].

## Study design and supplement administration

One study [33] employed an observer-blinded randomized placebo-controlled trial. Of the remaining 22 studies, one reported blinding only for the participants [38], whilst the others reported a randomized double-blind design with a placebo and a polyphenol supplement as the treatment arm. Various dietary polyphenols were investigated in the studies, with the main groups being Isoflavones [39, 54, 58], Flavonoids [30–32, 38, 40, 50–52, 56, 57, 60], Stilbenes (resveratrol) [49, 53, 55, 59]. Spearmint extract including rosmarinic acid was used in one study [29], as were curcumin [47], walnuts [33] and blueberry powder without polyphenolic content reported [48].

**Methodological quality of studies.** None of the reviewed studies scored less than 2 with a mean Jadad score of 4.08 ± 0.84. Of the 23 included studies, eight received a high score of 5/5, where all criteria were met. Ten studies received a score of 4/5, four studies scored 3/5 and 1 study scored 2/5. Overall, the quality was deemed to be good to excellent (see Table 3).

**Table 3. Quality assessment of studies using Jadad scale.**

| Studies | Items of Jadad scale | | | | | |
|---|---|---|---|---|---|---|
| | Randomization mentioned | Randomization appropriate | Blinding mentioned | Blinding appropriate | An account of all participants | Total score |
| [32] Ahles et al. 2020 | + | + | + | - | + | 4 |
| [31] Alharbi et al. (2016) | + | + | + | + | - | 4 |
| [49] Anton et al. (2018) | + | + | + | + | - | 4 |
| [50] Boespflug et al. (2017) | + | - | + | + | + | 4 |
| [51] Bondonno et al. (2020) | + | + | - | + | + | 4 |
| [30] Bowtell et al. (2017) | + | - | + | + | - | 3 |
| [52] Cook et al. (2020) | + | - | + | - | - | 2 |
| [47] Cox et al. (2014) | + | - | + | + | + | 4 |
| [38] Dodd et al. (2019) | + | + | - | + | - | 3 |
| [53] Evans et al. (2017) | + | + | + | + | + | 5 |
| [39] Fournier et al. (2007) | + | - | + | + | + | 4 |
| [54] Henderson et al. (2012) | + | + | + | + | + | 5 |
| [29] Herrlinger et al. (2018) | + | - | + | - | + | 3 |
| [55] Huhn et al. (2018) | + | + | + | + | + | 5 |
| [56] Igwea et al. (2020) | + | + | - | - | + | 3 |
| [57] Keane et al. (2016) | + | - | + | + | + | 4 |
| [40] Kennedy et al. (2015) | + | + | + | + | + | 5 |
| [58] Kreijkamp-Kaspers et al. (2004) | + | + | + | + | + | 5 |
| [48] Krikorian et al. (2022) | + | - | + | + | + | 4 |
| [33] Sala-Vila et al. (2020) | + | + | + | + | + | 5 |
| [59] Wong et al. (2013) | + | + | + | + | + | 5 |
| [60] Yahya et al. (2017) | + | + | + | + | + | 5 |
| [61] You et al. (2021) | + | + | + | + | - | 4 |

A score of 5 denotes the higher score to reach in Jadad scale

**Effect of polyphenol supplementation on executive functions.** Regarding acute polyphenols supplementation ($\leq$ 1-week), a significant effect was found for EFs on the Digit Vigilance and Serial 3 Tasks following curcumin supplementation [47]. A significant effect on performance speed across four EF tasks following flavonoid supplementation at 800 mg but not 1600 mg was reported by Kennedy et al. [40]. The same study also reported an interaction between treatment and testing occasion for spatial span scores at the 800 mg but not the 1600 mg dosage. However, the overall effect size for this study was nonsignificant (See meta-analysis section below). Chronic supplementation of 90 days with 900 mg per day of spearmint extract was associated with improved spatial and numeric working memory performance in one study, although a dosage of 600 mg per day showed no effect [29]. A four-week curcumin supplementation [47] was also reported to have a significant effect on serial 3 task. Chronic resveratrol supplementation [53, 59] (for 12 and 14 weeks at 75 mg and 150 per day respectively) was associated with improved performance on only one of eight Stroop test measures reported [59] and only trail making task B (TMT B) [53], although overall effect size for these studies were significant (See meta-analysis section below). Finally, daily administration of blueberry powder over 12 weeks was associated with enhanced performance on one condition of the Controlled Word Association Test, but the overall effect size for the study was not significant [48]. The remaining 17 studies reported no significant effects on individual EF performance measures and yielded no significant effect sizes overall (see below).

## Meta-analyses

**Level one meta-analysis of all primary studies.** The random-effects model for EF results for the 23 primary studies in Table 2 produced a nonsignificant summary effect size ($g = 0.076$, $z = 1.586$, *ns.*; $Q$ ($df = 22$) = 23.33, ns., $I^2 = 5.699$, Tau squared = 0.003). Fig 2 summarises the results of this meta-analysis.

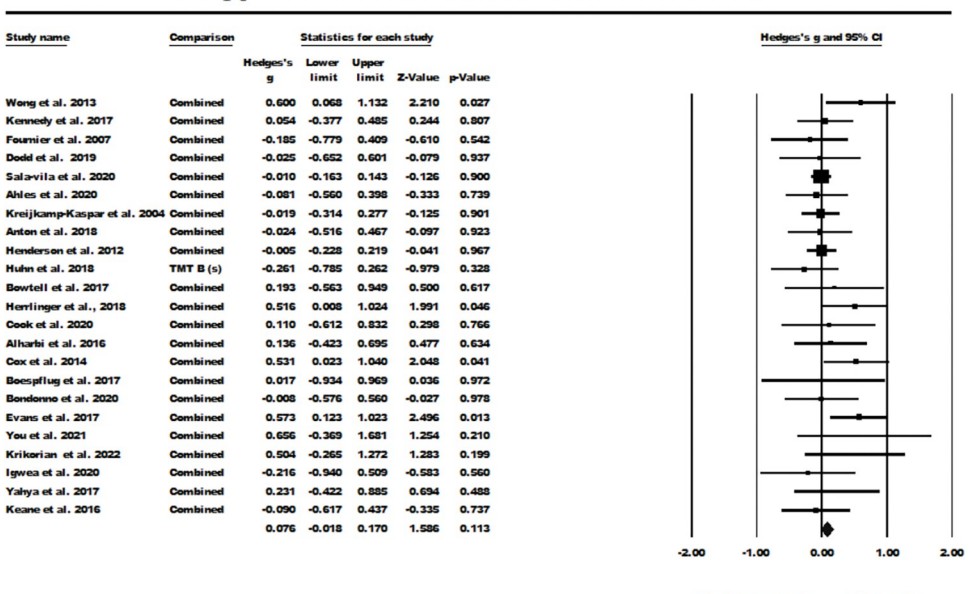

**Fig 2. Forest plot of studies investigating the effect of polyphenol supplementation on executive functions for all the primary studies.**

**Fig 3. Funnel plot of standard errors by hedges g showing both observed and adjusted effect summary for all the primary studies.**

About publication bias, Duval and Tweedie's trim and fill procedure trimmed one study with a positive effect, but only adjusted the summary effect slightly (adjusted $g$ = 0.072) (Fig 3). Both Kendal tau with continuity correction (Tau = 0.206 *ns.*) and Egger's regression procedure ($t$ (21) = 1.705, *ns.*) were nonsignificant when evaluated as two-tailed. It was concluded that there was no evidence of significant publication bias in this meta-analysis.

## Level two meta-analyses differentiating type of trial

**Between participant design RCT (BTW-P RCTs).**   As the 15 RCTs in the sample had used a variety of polyphenol supplements and treatment regimes, a random-effects model was used due to the distribution of true effect sizes which might be expected. The results of this model were nonsignificant ($g$ = 0.096, $z$ = 1.479, *ns.*; $Q$ ($df$ = 14) = 18.304, *ns.*, $I^2$ = 23.415, Tau squared = 0.013). Fig 4 summarises the results of this meta-analysis.

**Crossover trials (CO).**   As the eight crossover trials in the sample had used a variety of polyphenol supplements and treatment regimes, a random-effects model was used due to the distribution of true effect sizes which might be expected. The results of this model were non-significant ($g$ = 0.089, $z$ = 0.870, *ns.*; $Q$ ($df$ = 7) = 4.962, ns., $I^2$ = 0.000, Tau squared = 0.000). This value of tau-squared indicates that the random effects and fixed-effect models were equivalent for this sample [43]. Fig 4 also summarises the results of this analysis.

**Level three meta-analyses by polyphenol type.**   These meta-analyses were performed where it was possible to group the primary studies by polyphenol type. Studies that did not report explicitly the type or polyphenol's group of their supplementation were grouped together as "unclassifiable" and not included in Level Three meta-analyses. As polyphenol type and study design were controlled in these analyses, it was considered that the *a priori* likelihood of a distribution of true effect sizes being sampled was sufficiently diminished for fixed-effect models to be used in the first instance [43].

**Flavonoids.**   Twelve studies had administered flavonoids, with five BTW-P RCTs and seven crossover designs. For the BTW-P RCTs the results were nonsignificant ($g$ = 0.111, $z$ = 0.706, *ns.*; $Q$ ($df$ = 4) = 1.918, *ns.*, $I^2$ = 0.000, Tau squared = 0.000). The results of this analysis are shown in Fig 5. The fixed-effect model results for the seven crossover studies

## Subgroup analysis by study design

| Group by Study design | Study name | Comparison | Statistics for each study | | | | | Hedges's g and 95% CI |
|---|---|---|---|---|---|---|---|---|
| | | | Hedges's g | Lower limit | Upper limit | Z-Value | p-Value | |
| | Kennedy et al. 2017 | Combined | 0.054 | -0.377 | 0.485 | 0.244 | 0.807 | |
| | | | 0.054 | -0.377 | 0.485 | 0.244 | 0.807 | |
| BTW-P RCT | Ahles et al. 2020 | Combined | -0.081 | -0.560 | 0.398 | -0.333 | 0.739 | |
| BTW-P RCT | Anton et al. 2018 | Combined | -0.024 | -0.516 | 0.467 | -0.097 | 0.923 | |
| BTW-P RCT | Boespflug et al. 2017 | Combined | 0.017 | -0.934 | 0.969 | 0.036 | 0.972 | |
| BTW-P RCT | Bowtell et al. 2017 | Combined | 0.193 | -0.563 | 0.949 | 0.500 | 0.617 | |
| BTW-P RCT | Cox et al. 2014 | Combined | 0.531 | 0.023 | 1.040 | 2.048 | 0.041 | |
| BTW-P RCT | Evans et al. 2017 | Combined | 0.573 | 0.123 | 1.023 | 2.496 | 0.013 | |
| BTW-P RCT | Fournier et al. 2007 | Combined | -0.185 | -0.779 | 0.409 | -0.610 | 0.542 | |
| BTW-P RCT | Henderson et al. 2012 | Combined | -0.005 | -0.228 | 0.219 | -0.041 | 0.967 | |
| BTW-P RCT | Herrlinger et al., 2018 | Combined | 0.516 | 0.008 | 1.024 | 1.991 | 0.046 | |
| BTW-P RCT | Huhn et al. 2018 | TMT B (s) | -0.261 | -0.785 | 0.262 | -0.979 | 0.328 | |
| BTW-P RCT | Kreijkamp-Kaspar et al. 2004 | Combined | -0.019 | -0.314 | 0.277 | -0.125 | 0.901 | |
| BTW-P RCT | Krikorian et al. 2022 | Combined | 0.504 | -0.265 | 1.272 | 1.283 | 0.199 | |
| BTW-P RCT | Sala-vila et al. 2020 | Combined | -0.010 | -0.163 | 0.143 | -0.126 | 0.900 | |
| BTW-P RCT | Yahya et al. 2017 | Combined | 0.231 | -0.422 | 0.885 | 0.694 | 0.488 | |
| BTW-P RCT | You et al. 2021 | Combined | 0.656 | -0.369 | 1.681 | 1.254 | 0.210 | |
| BTW-P RCT | | | 0.096 | -0.031 | 0.224 | 1.479 | 0.139 | |
| CO | Alharbi et al. 2016 | Combined | 0.136 | -0.423 | 0.695 | 0.477 | 0.634 | |
| CO | Bondonno et al. 2020 | Combined | -0.008 | -0.576 | 0.560 | -0.027 | 0.978 | |
| CO | Cook et al. 2020 | Combined | 0.110 | -0.612 | 0.832 | 0.298 | 0.766 | |
| CO | Dodd et al. 2019 | Combined | -0.025 | -0.652 | 0.601 | -0.079 | 0.937 | |
| CO | Igwea et al. 2020 | Combined | -0.216 | -0.940 | 0.509 | -0.583 | 0.560 | |
| CO | Keane et al. 2016 | Combined | -0.090 | -0.617 | 0.437 | -0.335 | 0.737 | |
| CO | Wong et al. 2013 | Combined | 0.600 | 0.068 | 1.132 | 2.210 | 0.027 | |
| CO | | | 0.098 | -0.127 | 0.323 | 0.854 | 0.393 | |

Polyphenols          Placebo

**Fig 4. Forest plot of the subgroup meta-analysis for the between participants RCTs (BTW-P RCT) and crossover studies (CO) using random effect model.**

administering flavonoids were also nonsignificant ($g$ = 0.007, $z$ = 0.01, ns.; $Q$ ($df$ = 6) = 0.809, $ns.$, $I^2$ = 0.000, Tau squared = 0.000). The results of this analysis are also shown in Fig 5.

**Isoflavone.** Three primary studies (all BTW-P RCTs) had administered Isoflavone. The fixed-effect model for this sub-group was nonsignificant ($g$ = -0.024, $z$ = -0.278, $ns.$ $Q$ ($df$ = 2) = 0.317, $ns.$, $I^2$ = 0.000, Tau squared = 0.000). Fig 6 summarises the results of this analysis.

**Resveratrol.** Three of the four studies where resveratrol was administered were RCTs with between-participant designs, whilst one [59] was a crossover trial which had yielded a significant effect size (see Fig 2). For the BTW-P RCTs the results of the fixed-effect model were nonsignificant ($g$ = 0.139, $z$ = 0.976, $ns.$ $Q$ ($df$ = 2) = 6.246, $P$ = 0.044, $I^2$ = 67.978, Tau squared = 0.131). As heterogeneity was a problem in the fixed-effect model, a random-effects model was used but was also nonsignificant ($g$ = 0.109, $z$ = 0.431, $ns.$). However, the low power

## Subgroup analysis for the studies administering flavonoids

| Group by Study design | Study name | Comparison | Statistics for each study | | | | | Hedges's g and 95% CI |
|---|---|---|---|---|---|---|---|---|
| | | | Hedges's g | Lower limit | Upper limit | Z-Value | p-Value | |
| BTW-P RCT | Ahles et al. 2020 | Combined | -0.081 | -0.560 | 0.398 | -0.333 | 0.739 | |
| BTW-P RCT | Boespflug et al. 2017 | Combined | 0.017 | -0.934 | 0.969 | 0.036 | 0.972 | |
| BTW-P RCT | Bowtell et al. 2017 | Combined | 0.193 | -0.563 | 0.949 | 0.500 | 0.617 | |
| BTW-P RCT | Yahya et al. 2017 | Combined | 0.231 | -0.422 | 0.885 | 0.694 | 0.488 | |
| BTW-P RCT | You et al. 2021 | Combined | 0.656 | -0.369 | 1.681 | 1.254 | 0.210 | |
| BTW-P RCT | | | 0.111 | -0.197 | 0.420 | 0.706 | 0.480 | |
| CO | Alharbi et al. 2016 | Combined | 0.136 | -0.423 | 0.695 | 0.477 | 0.634 | |
| CO | Bondonno et al. 2020 | Combined | -0.008 | -0.576 | 0.560 | -0.027 | 0.978 | |
| CO | Cook et al. 2020 | Combined | 0.110 | -0.612 | 0.832 | 0.298 | 0.766 | |
| CO | Dodd et al. 2019 | Combined | -0.025 | -0.652 | 0.601 | -0.079 | 0.937 | |
| CO | Igwea et al. 2020 | Combined | -0.216 | -0.940 | 0.509 | -0.583 | 0.560 | |
| CO | Keane et al. 2016 | Combined | -0.080 | -0.607 | 0.446 | -0.299 | 0.765 | |
| CO | Kennedy et al. 2017 | Combined | 0.054 | -0.377 | 0.485 | 0.244 | 0.807 | |
| CO | | | 0.007 | -0.208 | 0.222 | 0.061 | 0.951 | |

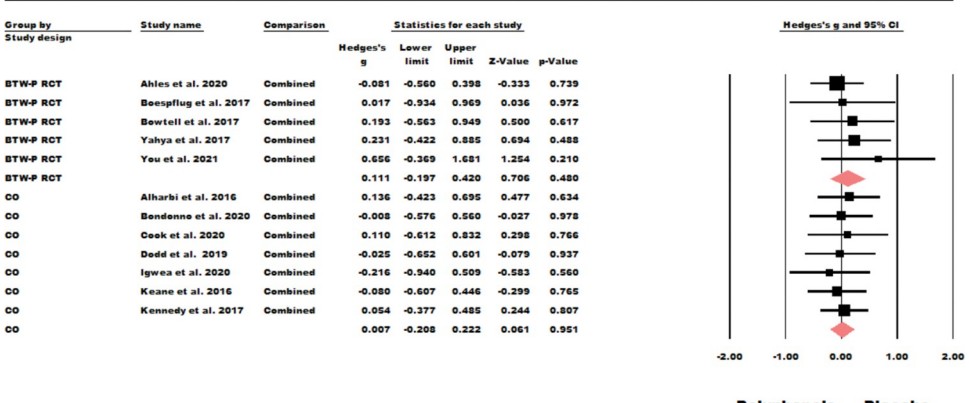

Polyphenols          Placebo

**Fig 5. Sub-group meta-analysis for the between participants RCTs (BTW-P RCT) and crossover studies administering flavonoids.**

### Sub-group analysis for the studies administering isoflavone

| Study name | Comparison | Statistics for each study | | | | | Std diff in means and 95% CI |
|---|---|---|---|---|---|---|---|
| | | Std diff in means | Lower limit | Upper limit | Z-Value | p-Value | |
| Fournier et al. 2007 | Combined | -0.189 | -0.795 | 0.416 | -0.614 | 0.539 | |
| Kreijkamp-Kaspar et al. 2004 | Combined | -0.019 | -0.316 | 0.278 | -0.125 | 0.901 | |
| Henderson et al. 2012 | Combined | -0.005 | -0.229 | 0.219 | -0.041 | 0.967 | |
| | | -0.024 | -0.196 | 0.147 | -0.278 | 0.781 | |

Polyphenols    Placebo

**Fig 6. Sub-group meta-analysis for the studies which administered isoflavone.**

of this random-effects model to accommodate a distribution of effect sizes should be noted [43]. Figs 7 and 8 summarises the results of fixed and random effect models tested for RCTs administrating resveratrol.

## Studies not included in level three meta-analyses

As previously noted, Wong et al. [59] yielded a positive effect size in the Level One meta-analysis following resveratrol administration, but this study was not included in the Level Three meta-analysis for resveratrol on the basis of study design. Four primary studies in Table 1 and Fig 2 were not included in Level Three meta-analyses because they could not be grouped with other studies based on the polyphenol administered [29, 33, 47, 48]. Two of these studies yielded significant effect sizes for EF following administration of rosmarinic acid [29] and curcumin [47], respectively.

## Sensitivity analyses

Sensitivity analyses were conducted on the Level Two meta-analyses previously reported, with each analysis being repeated with each trial in turn being omitted on each re-run (i.e., the analysis was repeated with n– 1 trials each time) (See S2 Table). None of the omissions resulted in a significant summary effect for either the RCTs or crossover trials.

### Fixed effect subgroup analysis for resveratrol

| Study name | Comparison | Statistics for each study | | | | | Hedges's g and 95% CI |
|---|---|---|---|---|---|---|---|
| | | Hedges's g | Lower limit | Z-Value | Upper limit | p-Value | |
| Anton et al. 2018 | Combined | -0.024 | -0.516 | -0.097 | 0.467 | 0.923 | |
| Huhn et al. 2018 | TMT B (s) | -0.261 | -0.785 | -0.979 | 0.262 | 0.328 | |
| Evans et al. 2017 | Combined | 0.573 | 0.123 | 2.496 | 1.023 | 0.013 | |
| | | 0.139 | -0.141 | 0.976 | 0.420 | 0.329 | |

Polyphenols    Placebo

**Fig 7. Sub-group meta-analysis for the between participants RCTs studies which administered resveratrol using fixed effect model.**

**Random effect subgroup analysis for resveratrol**

| Study name | Comparison | Statistics for each study | | | | | Hedges's g and 95% CI |
|---|---|---|---|---|---|---|---|
| | | Hedges's g | Lower limit | Z-Value | Upper limit | p-Value | |
| Anton et al. 2018 | Combined | -0.024 | -0.516 | -0.097 | 0.467 | 0.923 | |
| Huhn et al. 2018 | TMT B (s) | -0.261 | -0.785 | -0.979 | 0.262 | 0.328 | |
| Evans et al. 2017 | Combined | 0.573 | 0.123 | 2.496 | 1.023 | 0.013 | |
| | | 0.109 | -0.388 | 0.431 | 0.607 | 0.666 | |

-2.00  -1.00  0.00  1.00  2.00

Polyphenols          Placebo

**Fig 8. Sub-group meta-analysis for the between participants RCTs studies which administered resveratrol using random effect model.**

## Discussion

The present study investigated the effects of polyphenol supplementation on EFs in an obese/overweight population. All the meta-analyses yielded nonsignificant summary effects on EF performance, with there being no evidence to suggest the presence of publication bias. Individual primary studies showed significant overall effect sizes for curcumin [47] and phenolic acid [29], but these were the only studies to have used these respective supplements. Two further studies yielded significant effect sizes for resveratrol supplementation [53, 59], but sub-group analysis with the other two studies in the sample using resveratrol supplementation did not yield a significant summary effect for this polyphenol. Although Kennedy et al. [40] yielded an overall nonsignificant effect size in the meta-analysis, Table 2 shows that three measures were reported to have shown a significant positive effect following flavonoid supplementation. Similarly, Krikorian et al. [48] yielded a nonsignificant effect size for blueberry supplementation despite a significant effect having been reported on one section of the Controlled Oral Word Association Test.

The putative basis for polyphenols having a positive effect on cognition has included several mechanisms such as their antioxidant/anti-inflammatory effect [9, 27], their potential to improve blood pressure [32, 52], to enhance cerebral blood flow [40], and to increase resting-state cerebral perfusion [30]. Whilst all of these could potentially contribute to beneficial effects for EFs, and for other aspects of cognition, the findings of this present review provide some grounds to suggest that the age of participants who are obese/overweight could be contributing to the mediation of any beneficial effects for EFs. In the following comparisons of the ages of participants across studies, it is relevant to note the mean age of 62.92 years (SD = 8.06 years) for all participants across the 23 primary studies.

Chronic resveratrol supplementation was utilised by Anton et al. [49] and Huhn et al. [55] at a relatively high dose (> 200 mg per day) without yielding a significant effect in either case (Fig 2). However, the lower resveratrol doses administered by Wong et al. [59] and Evans et al. [53] can be seen in the same two figures to have yielded significant effects. These latter two studies both recruited younger participants (mean age approximately 61 years in both cases) compared to Anton et al. and Huhn et al. (mean ages approximately 73 and 68 years, respectively). Additionally, the significant effect observed with phenolic acid [29] was also in a younger sample (mean age approximately 58 years). When considering the two studies which did not yield significant overall effect sizes, but which did report at least one significant positive effect for flavonoid [40] and blueberry [48] supplementation, respectively, both had recruited

participants with mean ages between 55 and approximately 59 years. Three other studies using blueberry supplementation, all of which reported no significant effects, had older participant samples with mean ages between approximately 67 and 80 years [30, 38, 50]. These three studies all reported the flavonoid content of their supplementation, whereas Krikorian et al. [48] did not confirm the presence of flavonoid content in their blueberry powder preparation.

For flavonoid supplementation excluding blueberry preparations, no clear picture emerges concerning the relationship between participant's age and effect on EFs, with four studies [31, 32, 56, 61] reporting no significant effects following flavonoid administration to samples with younger mean ages than Kennedy et al. [40]. It must finally be noted that Cox et al. [47] yielded a significant effect size for curcumin from participant groups with mean ages in excess of 67 years. In summary, for resveratrol and blueberry preparations the relationship between the age of participants and effects on EFs should be investigated further in future trials recruiting from populations vulnerable to cognitive impairment. For other forms of polyphenol supplementation with this population, the question of age remains relevant in general terms given the less clear picture emerging from the present review.

Although Curcumin [47] was shown to have a significant effect in an older population (mean age 68 years) it is difficult to say that this was solely attributable to polyphenols as the exact constitution of curcumin supplementation was not reported. Moreover, it is worth pointing out that another two studies, Sala vila et al. [33] and Krikorian et al. [48] used walnuts and blueberry supplementation, respectively, without report the precise concentration and composition of polyphenolics administered. Consequently, the presence and potential contributions of plant derived bio active compounds other than polyphenols, cannot be ruled out. Thus, due to the wide range of phenolic compounds that might vary from one plant food to another [62], it is necessary for future studies investigating the effect of polyphenols to provide information relating to the constituents of supplementation in order to facilitate recognition of useful polyphenol compound/ group for cognitive health.

As previously mentioned, although EFs and some of the cognitive tasks which draw upon them have been identified empirically [17–19] and by logical deduction based upon reviews of published work [14, 16], the designation of a task as being an EF task can also be dependent upon decisions by researchers, with the possibility that this is then perpetuated by other researchers in subsequent research. The empirical and logical justification cited here for specific tasks being associated with specific executive functions is restricted to a relatively limited range of tasks used in cognitive research, and the psychological literature does not offer a systematic guide to those tasks which may be regarded as EF tasks. It follows that conclusions about the effect of polyphenols on EFs are necessarily dependent upon decisions regarding which tasks are relevant. This present review has been explicit regarding task and data selection, in terms of the four criteria reported in the Study Selection sub-section. It is recommended that future researchers report their basis for claiming that any task is measuring EFs in their participants, thus adding clarity to the literature concerning dietary supplementation across the range of cognitive functions. Clarity would also be enhanced by the reporting of means and standard deviations for all participant groups for all tasks administered as opposed, for example, to summarising nonsignificant results in the text of Results sections. Future reviews and meta-analyses would be helped by the adoption of this recommendation.

The present review has several limitations. There was only a relatively small number of primary studies concerning overweight and/or obese participants which met the inclusion criteria, with consequent limitations for the sub-group analysis. Some of the primary studies also had relatively small sample sizes. Problems concerning the identification of tasks measuring EFs have been previously mentioned, and some studies did not report the phenolic content of their supplementation, thus limiting the conclusions to be drawn. Meta-analysis is a statistical

methodology which synthesises data from a sample of primary studies [42, 43]. The outcome of the syntheses represented by the present meta-analyses were nonsignificant, but the presence of some significant positive findings in the primary studies indicate that further research is justified regarding the effect of polyphenol supplementation on EFs in people at risk of cognitive impairment. As previously noted, the role of age in mediating the effects of polyphenols in this population should particularly be investigated in relation to resveratrol [49, 55, 58, 59] and blueberry [30, 38, 48, 50] supplementation. Such research would be important because of the fundamental role of EFs in people's ability to regulate their daily lives through such things as planning, organising, modifying actions and decision making [14, 16–19], and because of the association of EF deterioration with the development of MCI and Alzheimer's disease [21–23]. Research into the prevention of EF impairment which may retard the development of dementia and severe cognitive decline in those who are at-risk due to obesity, should remain an important research priority.

## Conclusion

The findings from the present systematic review and meta-analyses indicated a nonsignificant effect of polyphenols on EFs. Nonetheless, further research should consider investigating polyphenols supplementation in a younger population at risk of cognitive impairment. Furthermore, the exact constituent of supplementation needs to be described as this is necessary for the identification of the potential beneficial compounds for cognitive health. Although research in the area of EFs might be challenging due to the variability in the tasks utilised, it is important to investigate the effect of polyphenols in this domain as prevention of EF deterioration might counteract the development of both MCI and Alzheimer's disease in at risk population.

## Supporting information

**S1 Checklist. PISMA checklist.**
(DOCX)

**S1 Table. Systematic review un-published protocol.**
(DOCX)

**S2 Table. Sensitivity analysis.**
(DOCX)

**S3 Table. Meta-analysis raw-data.**
(PDF)

## Acknowledgments

The authors would like to acknowledge the support received concerning the meta-analytic strategy in this article from Professor Andrew Clegg and Professor Catrin Tudor-Smith from the Methodological Innovation, Development, Adaptation and Support (MIDAS) Theme of the NIHR Applied Research Collaboration North-West Coast.

## Author Contributions

**Conceptualization:** Sara Farag, Catherine Tsang, Philip N. Murphy.

**Data curation:** Sara Farag, Philip N. Murphy.

**Formal analysis:** Sara Farag, Philip N. Murphy.

**Methodology:** Sara Farag.

**Project administration:** Sara Farag, Catherine Tsang, Philip N. Murphy.

**Software:** Sara Farag.

**Supervision:** Catherine Tsang, Philip N. Murphy.

**Validation:** Catherine Tsang, Philip N. Murphy.

**Visualization:** Sara Farag, Catherine Tsang, Philip N. Murphy.

**Writing – original draft:** Sara Farag, Catherine Tsang, Philip N. Murphy.

**Writing – review & editing:** Sara Farag, Catherine Tsang, Philip N. Murphy.

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
