## [Decision Letter · Decision Letter 0]

15 Feb 2023

PONE-D-22-35380Effect of Polyphenol Supplementation on Executive Function in Overweight and Obese Adults: A Systematic Review and Meta-AnalysisPLOS ONE

Dear Dr. Farag,

Thank you for submitting your manuscript to PLOS ONE. After careful consideration, we feel that it has merit but does not fully meet PLOS ONE’s publication criteria as it currently stands. Therefore, we invite you to submit a revised version of the manuscript that addresses the points raised during the review process.

ACADEMIC EDITOR's comments:  The title does not accurately reflect the study design, leading to a misperception that only individuals with no cognitive impairments were included in the study. It should clearly convey the population being studied

The introduction needs to be thoroughly revised to be more organized and focused. It should provide a clear overview of the research question, the relevant literature, and goals of the research. The manuscript, while focusing on executive functions and their specific measures in both cognitively healthy participants and those with mild cognitive impairment, fails to describe these in detail.

The authors should present a clear and concise summary of both primary and secondary outcomes.

We look forward to receiving your revised manuscript.

Kind regards,

Liliana G Ciobanu

Academic Editor

PLOS ONE

Journal Requirements:

"This work was funded by Edge Hill University PhD Studentship."

"SF: This work was funded by Edge Hill University PhD Studentship.The funders had no role in study design, data collection and analysis, decision to publish, or preparation of the manuscript."

4. Please upload a new copy of Figures 1-3 as the detail is not clear. Please follow the link for more information: https://blogs.plos.org/plos/2019/06/looking-good-tips-for-creating-your-plos-figures-graphics/" https://blogs.plos.org/plos/2019/06/looking-good-tips-for-creating-your-plos-figures-graphics/

Reviewers' comments:

Reviewer's Responses to Questions

**Comments to the Author**

1. Is the manuscript technically sound, and do the data support the conclusions?

Reviewer #1: Partly

Reviewer #2: Partly

2. Has the statistical analysis been performed appropriately and rigorously? 

Reviewer #1: No

Reviewer #2: Yes

3. Have the authors made all data underlying the findings in their manuscript fully available?

Reviewer #1: Yes

Reviewer #2: Yes

4. Is the manuscript presented in an intelligible fashion and written in standard English?

Reviewer #1: Yes

Reviewer #2: Yes

5. Review Comments to the Author

Reviewer #1: This study is about the effects of polyphenol on EF in obesity and overweight. Because of loose inclusion criteria, obvious heterogeneity should exist, but Q statistic and I square both showed no heterogeneity, which may indicate the meta-analysis analytic strategy. Authors should further justify and validate of their meta-analysis strategy or give an reasonable interpretation regarding to heterogeneity.

Reviewer #2: Dear Editor

This aim of this review is to describe the Effect of Polyphenol Supplementation on Executive Function in Overweight and Obese Adults: A Systematic Review and Meta-Analysis. Seventeen trials including were identified and the results were analyzed using random effect model. The results show that a potential positive effect of polyphenol supplementation in a younger population at-risk of cognitive impairment and it is recommended to investigate this further in future studies. Moreover, the variability of the tasks used to examine executive functions as well as the adequate reporting of supplement’s phenolic composition is a limitation that future work should also consider

The topic is of interest. However, there are some minor concerns to be addressed.

- Please explain all abbreviations in the abstract and manuscript.

Abstracts

- The word Cochrane is not used to describe the Cochrane trials library, please correct it

- The search was performed more than 6 month ago so I think authors should update their search

- Abstract should be informative, did they have any language preference?

- Abstract should be informative, what were the method they applied for this research are there any sensitivity and publication bias? They use SMD or WMD? Please explain

- Keywords: are these keywords are Mesh terms? Word that serves as a keyword, as to the meaning of that condition must be a Mesh term

Introduction

The introduction is disorganized and does not follow a main goal.

Methods

-While material and methods section is the most important part of a systematic review and meta-analysis, the main pitfall of this manuscript is its method section. PRISMA checklist is a good guide for this purpose. I noted that the manuscript is not written based on the PRISMA checklist and Cochran handbook

- Indicate if a review protocol exists, if and where it can be accessed and, if available, provide registration information including registration number (no problem if it does not exist, just state)

-List and define all variables for which data were sought (e.g., PICOS, funding sources) and any assumptions and simplifications made.

-Why authors a priori used a random-effects model for their analysis? Where there any signs to expect significant heterogeneity from the studies included?

- Authors should include a paragraph where all the outcomes (primary and secondary) of this meta-analysis are clearly summarized. This way the reader can easily track down each outcome of interest.

- The search was performed more than 6 month ago so I think authors should update their search

- What are these key words? It is appears that they are suitable for just MEDLINE search, so what about other databases? I cannot search SCOPUS with these syntaxes. If the authors use different syntaxes, they should present them, if they don’t at least present keyword in plain form not in a specific database format.

- What about other source of potential included articles such as grey literatures? Did the authors searched this references?

- The type of clinical trial studies is different in each of the randomized, parallel, and crossover cases, and you cannot combine these studies; these studies must be separated or at least calculated for each subgroup study.

- Did the authors performed sensitivity analysis for included studies? I cannot find its explanation in the method section

- The quality assessment tool used was not the updated tool (Rob2) provided by the Cochrane collaboration group. I suggest use this new tool for this step. JADAD score is an off-date score

6. PLOS authors have the option to publish the peer review history of their article (what does this mean?). If published, this will include your full peer review and any attached files.

Reviewer #1: **Yes: **Yong Zhang

Reviewer #2: No

---

## [Author Response · Author response to Decision Letter 0]

30 Apr 2023

Dear Editor,

Re: PONE-D-22-35380

Thank you for the opportunity to submit our revised manuscript. We have listed below our responses to the points made by you as the academic editor, and by the reviewers.

ACADEMIC EDITOR's comments: 

The title does not accurately reflect the study design, leading to a misperception that only individuals with no cognitive impairments were included in the study. It should clearly convey the population being studied

Response. The title has now been revised to reflect more accurately the focus of the study. The new title is ‘Polyphenol supplementation and executive functioning in overweight and obese adults at risk of cognitive impairment: a systematic review and meta-analysis’. 

The introduction needs to be thoroughly revised to be more organized and focused. It should provide a clear overview of the research question, the relevant literature, and goals of the research. The manuscript, while focusing on executive functions and their specific measures in both cognitively healthy participants and those with mild cognitive impairment, fails to describe these in detail. The authors should present a clear and concise summary of both primary and secondary outcomes.

Response. The Introduction has been extensively re-written to reflect the concerns expressed by the academic editor

Notes 

The marked-up copy of the manuscript: When applicable, and in some instances when a whole section was changed ( such as in the case of the introduction), the font colour was made dark blue, other than that all changes have been highlighted as requested. Figures: All Figures have been corrected using PACE as suggested.

Financial disclosure: An updated financial disclosure statement has been added to the cover letter

Reviewers' comments:

Reviewer #1: This study is about the effects of polyphenol on EF in obesity and overweight. Because of loose inclusion criteria, obvious heterogeneity should exist, but Q statistic and I square both showed no heterogeneity, which may indicate the meta-analysis analytic strategy. Authors should further justify and validate of their meta-analysis strategy or give an reasonable interpretation regarding to heterogeneity.

Response. With regard to the choice of model, Borenstein et al [1 P. 84] state that “…the decision to use the random-effects model should be based on our understanding of whether or not all studies share a common effect size and not the outcome of a statistical test (especially since the test for heterogeneity often suffers from low power).” They go on to say [1 P. 85] “If the study effect sizes are seen as having been sampled from a distribution of effect sizes, then the random-effects model, which reflects this idea, is the logical one to use.” 

In keeping with the warnings of Borenstein et al., there were several reasons for not wishing to make an assumption concerning a common effect size in the Level One and Level Two meta-analyses. Firstly, the difference in type of polyphenol type administered, and the differences in dosing regimens used between studies made such an assumption questionable. Secondly, mild cognitive impairment was a feature of the participant population we wanted to include, due to the relationships between being overweight, neuroinflammation, neuronal damage, and consequent cognitive impairment. However, we considered it necessary to be open to the possibility that such impairments may be present to differing extents and may or may not be identified and reported by the primary studies. Consequently, we decided to avoid the assumption of a common effect size and chose the fail-safe option of a random-effects model. The revised draft does, however, use fixed-effect models in the Level 3 meta-analyses, and the rationale for this is explained in the article.

Reviewer #2: Dear Editor

This aim of this review is to describe the Effect of Polyphenol Supplementation on Executive Function in Overweight and Obese Adults: A Systematic Review and Meta-Analysis. Seventeen trials including were identified and the results were analysed using random effect model. The results show that a potential positive effect of polyphenol supplementation in a younger population at-risk of cognitive impairment and it is recommended to investigate this further in future studies. Moreover, the variability of the tasks used to examine executive functions as well as the adequate reporting of supplement’s phenolic composition is a limitation that future work should also consider

The topic is of interest. However, there are some minor concerns to be addressed.

- Please explain all abbreviations in the abstract and manuscript.

Response. All abbreviations were reviewed and explained as they first appear in the manuscript.

Abstracts

- The word Cochrane is not used to describe the Cochrane trials library, please correct it.

Response. The word Cochrane was corrected to Cochrane trials library

- The search was performed more than 6 month ago so I think authors should update their search.

Response. The search was updated, and 6 new studies have been identified that meet the inclusion criteria and were added to the other studies for meta-analysis. 

- Abstract should be informative, did they have any language preference?

Response. The inclusion criteria for the search were to include peer reviewed articles published in English. This information is now described in the abstract. 

- Abstract should be informative, what were the method they applied for this research are there any sensitivity and publication bias? They use SMD or WMD? Please explain.

Response. Hedges g was used as the effect size metric for all variables to avoid distortions arising from small sample sizes. The following methods for examining publication bias were examined, based upon their description by Borenstein et al [1]: Rosenthal’s Fail-safe N, funnel plots, the Begg and Mazumdar’s rank correlation test (Kendall’s S statistic P-Q) , Egger’s linear regression test, and Duval and Tweedie’s trim-and-fill test. In each case, it was clear that the rationale of the test was to identify if significant summary effects were the result of publication bias. Consequently, to conduct such analyses on meta-analytic results with nonsignificant summary effects would clearly be pointless and redundant. In this event, all our meta-analytic results were nonsignificant. However, our meta-analytic strategy sub-section explains our rationale regarding publication bias analyses.

 - Keywords: are these keywords are Mesh terms? Word that serves as a keyword, as to the meaning of that condition must be a Mesh term

Response. The keywords are Mesh terms. The keyword Obese was changed to Obesity. All keywords were checked using the following link: https://www.ncbi.nlm.nih.gov/mesh/?term=systematic+review

Introduction

The introduction is disorganized and does not follow a main goal.

Response. The introduction was revised and updated as advised.

Methods

-While material and methods section is the most important part of a systematic review and meta-analysis, the main pitfall of this manuscript is its method section. PRISMA checklist is a good guide for this purpose. I noted that the manuscript is not written based on the PRISMA checklist and Cochran handbook.

Response. The method section was revised, PRISMA checklist was followed and provided as a supplementary material to the Manuscript. In addition, we used a recent PLOSONE publication by Lei et al. (2022) titled” Non-pharmacological interventions on anxiety and depression in lung cancer patients’ informal caregivers: A systematic review and meta-analysis” was used as a guide to comply with both, the journal requirements and PRISMA checklist.

- Indicate if a review protocol exists, if and where it can be accessed and, if available, provide registration information including registration number (no problem if it does not exist, just state).

Response. A review protocol was not published and as advised; this was stated in the method section. The un-published research protocol can be found in the supplementary materials as S2 Table.

-List and define all variables for which data were sought (e.g., PICOS, funding sources) and any assumptions and simplifications made.

Response. The data were sought using PICOS and this is now presented as Table 1 in the manuscript.

-Why authors a priori used a random-effects model for their analysis? Where there any signs to expect significant heterogeneity from the studies included?

Response. The study effect sizes were sampled from a distribution of effect sizes which made the use of Random effects model the appropriate model to decide on well before conducting the analysis. Please refer to the updated Meta-analysis strategy in the method section and our response to reviewer #1 who shared a similar concern. 

- Authors should include a paragraph where all the outcomes (primary and secondary) of this meta-analysis are clearly summarized. This way the reader can easily track down each outcome of interest.

Response. The primary and the secondary outcomes were outlined at the last part of the introduction as advised.

- The search was performed more than 6 month ago so I think authors should update their search.

Response. The search was updated and 6 new studies were included for the final analysis.

- What are these key words? It is appears that they are suitable for just MEDLINE search, so what about other databases? I cannot search SCOPUS with these syntaxes. If the authors use different syntaxes, they should present them, if they don’t at least present keyword in plain form not in a specific database format.

Response. The included search terms were keywords that were used to conduct the search, and these are now presented in plain format. The methods used in their use and how they were combined are now described in the method section of the manuscript. 

- What about other source of potential included articles such as grey literatures? Did the authors searched this references?

Response. The grey literature was not searched. However, it is our experience that RCTs are not commonly published there

- The type of clinical trial studies is different in each of the randomized, parallel, and crossover cases, and you cannot combine these studies; these studies must be separated or at least calculated for each subgroup study.

Response. Base on a consultation we had with our external advisors, a separate sub-group analysis was conducted for the parallel and cross-over studies as described in the updated meta-analysis strategy and results section. However, we were advised to report the overall effect size for both types of these studies. It should also be noted that in following the guidance of Borenstein et al [1 see P227) concerning the use of the mean of comparison outcomes for studies reporting multiple comparisons based upon the same participants, we have avoided the risk of a distortion to our summary effect size from trials with parallel arms reporting comparisons with a common control (placebo) group.

- Did the authors performed sensitivity analysis for included studies? I cannot find its explanation in the method section.

Response. Yes. A sensitivity analysis was performed, and its results are outlined in the supplementary material as S1 Table. 

- The quality assessment tool used was not the updated tool (Rob2) provided by the Cochrane collaboration group. I suggest use this new tool for this step. JADAD score is an off-date score.

Response. Although Rob2 is a quality assessment tool advised by the Cochrane collaboration group, the Jadad scale, and other tools, are also validated tools that is still being used and according to PubMed, Jadad Scale was reported in 188 studies published in 2022. An example for studies that used Jadad scale is a recent PLOSONE publication by Cao et al. (2022) titled “Efficacy of Banxia Xiexin decoction for chronic atrophic gastritis: A systematic review and meta-analysis“. In addition, Lei et al. (2022) (described above) used a different tool which is the Physiotherapy Evidence Database (PEDro) scale. Given that the tool we used is a validated tool that is still being implemented and accepted in peer reviewed publications and that Rob2 is not the only tool to be used for quality assessment, we have not changed our used tool to ROB 2.

---

## [Editor Report · Decision Letter 1]

10 May 2023

Polyphenol supplementation and executive functioning in overweight and obese adults at risk of cognitive impairment: a systematic review and meta-analysis.

PONE-D-22-35380R1

Dear Sara Farag,

We’re pleased to inform you that your manuscript has been judged scientifically suitable for publication and will be formally accepted for publication once it meets all outstanding technical requirements.

Kind regards,

Liliana G Ciobanu

Academic Editor

PLOS ONE

---

## [Editor Report · Acceptance letter]

15 May 2023

PONE-D-22-35380R1 

‘Polyphenol supplementation and executive functioning in overweight and obese adults at risk of cognitive impairment: a systematic review and meta-analysis’. 

Dear Dr. Farag:

I'm pleased to inform you that your manuscript has been deemed suitable for publication in PLOS ONE. Congratulations! Your manuscript is now with our production department. 

Kind regards, 

on behalf of

Dr. Liliana G Ciobanu 

Academic Editor

PLOS ONE